# Moss-Derived Human Recombinant GAA Provides an Optimized Enzyme Uptake in Differentiated Human Muscle Cells of Pompe Disease

**DOI:** 10.3390/ijms21072642

**Published:** 2020-04-10

**Authors:** Stefan Hintze, Sarah Limmer, Paulina Dabrowska-Schlepp, Birgit Berg, Nicola Krieghoff, Andreas Busch, Andreas Schaaf, Peter Meinke, Benedikt Schoser

**Affiliations:** 1Friedrich-Baur-Institute, Department of Neurology, Ludwig-Maximilians-University Munich, 80336 Munich, Germany; stefan.hintze@med.uni-muenchen.de (S.H.); sarah.limmer@med.uni-muenchen.de (S.L.); peter.meinke@med.uni-muenchen.de (P.M.); 2Greenovation Biotech GmbH, 79108 Freiburg, Germany; PDabrowska@greenovation.com (P.D.-S.); BBerg@greenovation.com (B.B.); NKrieghoff@greenovation.com (N.K.); ABusch@greenovation.com (A.B.); ASchaaf@greenovation.com (A.S.)

**Keywords:** Pompe disease, enzyme replacement therapy, glycogen storage disease type II, moss-GAA

## Abstract

Pompe disease is an autosomal recessive lysosomal storage disorder (LSD) caused by deficiency of lysosomal acid alpha-glucosidase (GAA). The result of the GAA deficiency is a ubiquitous lysosomal and non-lysosomal accumulation of glycogen. The most affected tissues are heart, skeletal muscle, liver, and the nervous system. Replacement therapy with the currently approved enzyme relies on M6P-mediated endocytosis. However, therapeutic outcomes still leave room for improvement, especially with regard to skeletal muscles. We tested the uptake, activity, and effect on glucose metabolism of a non-phosphorylated recombinant human GAA produced in moss (moss-GAA). Three variants of moss-GAA differing in glycosylation pattern have been analyzed: two with terminal mannose residues in a paucimannosidic (Man3) or high-mannose (Man 5) configuration and one with terminal N-acetylglucosamine residues (GnGn). Compared to alglucosidase alfa the moss-GAA GnGn variant showed increased uptake in differentiated myotubes. Moreover, incubation of immortalized muscle cells of *Gaa*^−/−^ mice with moss-GAA GnGn led to similarly efficient clearance of accumulated glycogen as with alglucosidase alfa. These initial data suggest that M6P-residues might not always be necessary for the cellular uptake in enzyme replacement therapy (ERT) and indicate the potential of moss-GAA GnGn as novel alternative drug for targeting skeletal muscle in Pompe patients.

## 1. Introduction

Pompe disease (also termed glycogen storage disease type II; OMIM #232300) is caused by compound heterozygous or homozygous mutations of the *GAA* gene, which encodes the acid alpha-1,4-glucosidase (GAA) [1,2]. Mutations in *GAA* result in GAA enzyme deficiency, which is a lysosomal enzyme catalyzing the degradation of glycogen. Clinical symptoms of the severe infantile onset form (IOPD) are cardiomyopathy and muscular hypotonia, while in the later-onset forms (LOPD) the involvement of skeletal muscles is the predominant phenotype of this multisystemic spectrum disorder [3].

Since 2006, a licensed enzyme replacement therapy (ERT) is available for all Pompe patients: alglucosidase alfa, a recombinant human GAA (rhGAA) carrying mannose 6-phosphate (M6P) glycans. Alglucosidase alfa is taken up into cells via the cation independent M6P-receptor for direct delivery to the lysosomes by receptor mediated endocytosis [4]. Treatment with alglucosidase alfa leads to remarkable amelioration in cardiac muscle in most cases, but its therapeutic efficacy in skeletal muscle and other tissues is restricted. Results of long-term clinical studies show stabilization and/or improvement in walking distance and respiratory function, while weakness of axial skeletal muscles usually persists or further declines [5,6]. Due to the limited clinical efficacy of this ERT, there is an ongoing need for improvements of the administrated enzyme in terms of stability and delivery efficacy.

Like in the case of some other lysosomal storage disorders (LSDs), alglucosidase alfa is being produced in Chinese hamster ovary (CHO) cells to ensure a human-like glycosylation profile, including terminal M6P-rests. Several different modifications of the alglucosidase alfa, aimed to improve enzyme uptake have been proposed e.g.,: crafting of M6P analogues [7,8], increasing the number of M6P residues on the enzyme [9], co-administration with a small molecule pharmacological chaperone [10], or using a glycosylation-independent lysosomal targeting tag–insulin-like growth factor [11].

In recent years, several lysosomal enzymes for therapeutic use in ERT have been successfully produced using plant-based expression platforms [12,13,14,15] (see [16] for review). GAA has been effectively produced in several plant-hosts, e.g., in rice calli [17] and tobacco [18]. Plant-based systems in general profit from advantages like lower production costs, eliminated risk of contamination by zoonotic pathogens and improved process stability. In the case of moss, its haploid nature and the extraordinary high rate of homologous recombination-based DNA-repair enable straightforward and stable manipulation of its genome. Additionally, moss-made proteins exhibit a very homogenous glycosylation profile in contrast to a variable glyco-pattern of many CHO-derived biopharmaceuticals. However, plants are devoid of pathways responsible for phosphorylation and thus are not able to generate the M6P motif on the expressed proteins. Therefore, plant-made lysosomal enzymes were considered inadequate for treatments relying predominantly on ERT trafficking via M6P-receptors until recently. However, we and others could show uptake of plant-made enzymes with mannose- and *N*-acetylglucosamine-terminated glycans into lysosomes for example of alpha-galactosidase, which is being deployed in treatment of another LSD–Fabry disease [6,15,19].

In this study we tested the uptake and activity of moss-produced recombinant human GAA (moss-GAA) in murine (Pompe model mouse) as well as human (originating from Pompe patients) myoblasts and differentiated myotubes, in comparison to alglucosidase alfa.

## 2. Results

### 2.1. Production, Purification, and Characterization of moss-GAA Variants

The 110 kDa human GAA precursor was stably overexpressed in the moss *Physcomitrella patens* and secreted to the culture medium. The moss-GAA GnGn expressing moss strain was a glycoengineered variant devoid of plant-specific α-1,3-fucose and β-1,2-xylose residues on its *N*-glycans [20]. We generated two additional glyco-variants of GAA: moss-GAA Man5, expressed in a strain additionally devoid of β-1,2-*N*-acetylglucosaminyltransferase activity (GnTI knockout) [15] and moss-GAA Man3, produced by enzymatic removal of *N*-acetylglucosamine residues from the GnGn variant (Figure 1A; for details refer to materials and methods).

In SDS-PAGE all three glyco-variants of the moss-GAA: GnGn, Man5, and Man3 showed a single major band with a faster mobility than alglucosidase alfa (Figure 1B, left panel), reflecting the lower carbohydrate content in moss-GAA versions (with the lowest in Man3 variant). In western blot, all recombinant GAA proteins were detected by a polyclonal anti-human GAA antibody (Figure 1B, right panel). Comparable specific activity could be shown for all three variants and the marketed alglucosidase alfa (Figure 1C).

HILIC-HPLC-MS analysis revealed a highly homogenous glycosylation profile of all three moss-GAA variants with exclusively mannose or *N*-acetylglucosamine terminated sugars in sharp contrast to the heterogeneous profile of alglucosidase alfa (Figure 1D).

### 2.2. Uptake and Activity of rhGAA in Mouse Myoblasts

Immortalized *Gaa*^−/−^ mouse myoblasts were treated with all three moss-GAA variants, alglucosidase alfa, and buffer (untreated) for 24 h. After 24 h an enzyme activity assay was performed (Figure 2A). The GAA activity was severely reduced in untreated *Gaa*^−/−^ myoblasts compared to wildtype. Alglucosidase alfa treatment resulted in a highly significant increase of enzyme activity to wildtype levels. All moss-GAA variants showed a highly significant increase in activity over untreated myoblasts as well but reached only about a quarter of the wildtype activity. Moss-GAA GnGn showed the highest activity of the three tested moss-GAA variants. Immunofluorescence staining for GAA (Figure 2B) showed faint signals in alglucosidase alfa and moss-GAA GnGn treated cells but very little to no detectable signal in moss-GAA Man5 and Man3.

### 2.3. Uptake and Activity of rhGAA in Mouse Myotubes

As the target organ in the Pompe disease is the mature muscle, we tested the uptake of all rhGAA variants in differentiated myotubes. The immortalized *Gaa*^−/−^ mouse myoblasts were differentiated for six days and thereafter treated with all three moss-GAA variants, alglucosidase alfa, and buffer (untreated) for 24 h. After 24 h myotubes were isolated to avoid contamination of undifferentiated myoblasts, and an enzyme activity assay was performed (Figure 3A). As expected, the GAA activity was severely reduced in untreated *Gaa*^−/−^ myotubes compared to wildtype. All variants of rhGAA showed a significant increase of GAA activity, but contrary to results in undifferentiated myoblasts, none of the variants reached wildtype level. Alglucosidase alfa showed a lower increase of activity compared to moss-GAA; and amongst the tested moss-GAA variants, the GnGn type generated the biggest increase in enzyme activity. Immunofluorescence staining for GAA (Figure 3B) showed faint signals in moss-GAA GnGn and Man5 treated myotubes but very little to no detectable signal in moss-GAA Man3 and alglucosidase alfa.

### 2.4. Uptake and Activity of rhGAA in Human Myoblasts

Being aware that differences between murine and human cells might affect the outcome of our measurements, we tested the uptake in different primary human myoblasts from Pompe patients next (Table 1). Primary patient myoblasts were treated with all three moss-GAA variants, alglucosidase alfa, and buffer (untreated) for 24 h. After 24 h an enzyme activity assay was performed (Figure 4B). All three patient myoblast cultures showed reduced GAA activity compared to healthy controls–although not to the extent of diseased mouse myoblasts, which might be explained by the residual GAA enzyme levels in Pompe patients. Alglucosidase alfa showed an 8–13 times increased enzyme activity compared to healthy controls levels. Furthermore, all three moss-GAA variants did show a significant increase in enzyme activity in all three patient myoblast cultures, with the GnGn variant reaching the highest levels of 50–100% of healthy controls. Immunofluorescence staining for GAA (Figure 4A) showed strong signals in alglucosidase alfa and moss-GAA-GnGn treated myoblasts and weak signals in moss-GAA-Man5 and -Man3.

Periodic acid–Schiff (PAS)-staining to detect glycogen depositions (Figure 4C) showed massive accumulation in untreated cells and a complete clearance of these accumulations following alglucosidase alfa and moss-GAA GnGn treatment. Both moss-GAA Man5 and Man3 variants also led to reduced glycogen accumulations.

### 2.5. Uptake and Activity of rhGAA in Human Myotubes

Next, we tested the uptake of all rhGAA variants in differentiated human patient myotubes. Myoblasts were differentiated for six days, thereafter, treated with all three moss-GAA variants, alglucosidase alfa, and buffer (untreated) for 24 h. After 24 h myotubes were isolated to avoid contamination of undifferentiated myoblasts, and an enzyme activity assay was performed (Figure 5B). The GAA activity was severely reduced in all patient-derived myotubes compared to a healthy control. The enzyme activity increased only in patient 4 cell line following alglucosidase alfa treatment, whereas treatment with moss-GAA Man5 resulted in an increased activity in patient 1 cells and moss-GAA GnGn treatment resulted in an increased activity in patients 1 and 4. We did not test moss-GAA-Man3 due to the amount of differentiated myotubes needed and the lesser improvements seen with this variant in previous experiments.

Immunofluorescence staining for GAA in myotubes (Figure 5A) detected faint signals only following moss-GAA GnGn and Man5 treatment. PAS-staining in patient-myotubes (Figure 5C) did show accumulation in untreated cells and only treatment with moss-GAA GnGn led to reduction of these depositions.

### 2.6. Metabolic Measurements

To investigate the effect of the recombinant GAA versions on metabolic performance of Pompe cells we performed measurements of the glycolysis using a Seahorse XFp analyzer (Figure 6). We could show previously a reduced glycolysis in murine and human *Gaa*^−/−^ /Pompe myoblasts which is partially rescued by alglucosidase alfa treatment [21]. Due to the limited size of the plate wells, measurements were performed with myoblasts only. In immortalized *Gaa*^−/−^ mouse myoblasts, treatment with alglucosidase alfa as well as moss-GAA GnGn for 24 h resulted in increased glycolysis rate while in case of moss-GAA Man5 and Man3 no significant change could be observed (Figure 6A). We performed parallel tests with human primary patient myoblasts. Similarly, to mouse myoblasts, treatment with alglucosidase alfa and moss-GAA GnGn led to an increased glycolysis rate (Figure 6B).

## 3. Discussion

In the past, evaluation of the cellular uptake of recombinant versions of GAA, as well as other LSD enzymes, has been frequently performed using immortalized patients’ fibroblast lines, mainly because they were easily obtainable via skin biopsies. It is possible that this focus on fibroblasts, which are abundant of M6P receptors, as cellular models of lysosomal diseases, introduced a certain bias towards the development of M6P-terminated enzymes [22]. Moreover, it is important to point out that fibroblasts do not represent the target cell type of the Pompe disease. In recent studies, the use of Pompe myoblasts for cellular uptake tests has been reported [8]. Nevertheless, myoblasts as progenitor cells can still potentially differ (in e.g., receptor type and their distribution in the cell membrane) from the fully differentiated muscle, which is the actual cellular target of the recombinant GAA. Being aware of these shortcomings, we tested the cellular uptake of moss-GAA variants in disease relevant myoblast and myotubes cell lines.

Testing the uptake and activity of three moss-GAA glyco-variants and comparing them with alglucosidase alfa in immortalized *Gaa*^−/−^ mouse myoblasts and myotubes revealed that in general the uptake into myotubes was less efficient than in myoblasts. While alglucosidase alfa increased enzyme activity to wildtype levels in myoblasts, only very modest activity elevation was observed in myotubes. Of the tested moss-GAAs the GnGn variant performed best in both cell lines and was performing better in myotubes than alglucosidase alfa. Immunofluorescence staining confirmed results of the GAA activity assay in both cell lines, as the detectable signal correlated with the measured activity. This indicates that the enzyme with terminal *N*-acetlyglucosamines is more efficiently taken up in mouse myotubes than the M6P-terminated one. 

As mouse and human cells might basically differ in gene expression and the immortalization process of the mouse cells can have an additional effect, we proceeded to perform the same tests in primary human patient cultures. Overall, we could confirm the results obtained with mouse cells. The only difference between human and mouse cells was generally higher enzyme activity in human myoblasts following treatment, which could indicate either better uptake or lower GAA levels and activity in healthy human myoblasts. The alglucosidase alfa activity reached about 10-times the wildtype level and moss-GAA GnGn reached wildtype level in myoblasts. There were, however, differences between patients suggesting that there are factors affecting the uptake in each individual. This fits well with the observation that alglucosidase alfa treatment has variable affects in patients living with Pompe disease [5,23,24,25]. These results were additionally backed up by the PAS-staining, which clearly showed the best reduction of glycogen deposits following the treatment with moss-GAA GnGn especially in case of myotubes, where exclusively the GnGn variant led to reduction of glycogen accumulations.

It is known that the GAA mutations negatively influence the metabolic functions of the cells [6,26]. We decided to test all moss-GAA variants in comparison to alglucosidase alfa for rescue of reduced glycolysis using the Agilent Seahorse technology, providing real-time, quantitative data on metabolic performance of Pompe myoblasts pre- and post-rhGAA treatment. Treatment with both alglucosidase alfa and moss-GAA GnGn led to significantly higher glycolysis than in the untreated control and thus was in line with previously acquired results. In human primary myoblasts, we measured clearly lower glycolysis in Pompe patients than in healthy controls. In this cell line, treatment with alglucosidase alfa and moss-GAA GnGn also indicated an improved metabolic performance.

The overall best performance of the GnGn glyco-variant was an unexpected finding. For another moss-derived lysosomal enzyme–alpha galactosidase (moss-aGal), we could show that the cellular uptake is predominantly mediated via the mannose receptor (MR) [15]. Accordingly, the variant of moss-aGal with paucomannosidic (Man3) configuration proved to be the most effective in cellular uptake as well as in the mouse Fabry model. Assuming the similar MR involvement in the endocytosis of moss-GAAs, one would expect a higher uptake rate with mannose-terminated Man3 and Man5 variants than with the GnGn type, although the N-acetylglucosamine glycans are also known to be recognized by the MR [27]. Moreover, in an uptake inhibition test with mannan and mannose-6-phosphate, we could observe a clear uptake inhibition of alglucosidase alfa with M6P, but mannan did not lead to full constraint of moss-GAAs internalization. These results suggest an involvement of an alternative receptor in the endocytosis of moss-derived GAA. Delivery pathways, which are mediated independently of M6P and MR are known [28,29] and several receptors have been proposed to play a role in cellular as well as lysosomal uptake in LSDs. Most of them are type I transmembrane proteins. We are currently planning further studies to identify the putative receptor, responsible for the preferential uptake of moss-GAA GnGn variant. Discovery of such an alternative internalization mechanism could open a way for new approaches in ERTs.

Our study demonstrated potential therapeutic effect of moss-derived human GAA, especially in treatment of skeletal muscle pathology in Pompe disease. Moss-GAA GnGn showed superiority over the current standard ERT enzyme: alglucosidase alfa, in cell uptake studies and efficacy in clearance of glycogen accumulations in Pompe patients’ myotubes. Considering the organismal level, it remains to be shown whether our results can be confirmed in the Pompe mouse model, as other tissues/organs could take up moss-GAA, and thus impact on muscle delivery.

## 4. Materials and Methods

### 4.1. Culture of Immortalized Mouse Cells

Immortalized *Gaa*^−/−^ mouse myoblasts cell line LA5 (gained from a *Gaa*^−/−^ mouse, [30]) were kindly provided by Nina Raben, National Institutes of Health, Division of Muscle Biology, USA. The myoblasts were cultured in DMEM supplemented with 20% FCS, 10% horse serum, GlutaMax, 1% Chick embryo extract, IFN-γ and 40 U/mL penicillin, 0.04 mg/mL streptomycin in an incubator at 33 °C with 5% CO_2_. All reagents were from ThermoFisher Germany, and Sigma-Aldrich, Germany.

Differentiation was started by adding differentiation medium (DMEM, 2% horse serum, GlutaMax, 0.5% chick embryo extract, 40 U/mL penicillin, and 0.04 mg/mL streptomycin) to 70% confluent cells at 37 °C with 5% CO_2_ over seven days.

### 4.2. Patients and Controls

Human control and patient materials were obtained with written informed consent of the respective donor from the Muscle Tissue Culture Collection (MTCC) at the Friedrich Baur Institute (Department of Neurology, Ludwig Maximilians University, Munich, Germany) (Table 1). Ethical approval for this study was obtained from the ethical review committee at the Ludwig Maximilians University, Munich, Germany (IRB-no. reference 45-14, year 2014).

### 4.3. Culture of Primary Skeletal Muscle Cells

Primary human myoblasts were cultured in Skeletal Muscle Growth Medium supplemented with SkMC Supplement (PELOBiotech, Pelobiotech GmbH, Martinried, Germany), GlutaMax and 40 U/mL Penicillin, 0.04 mg/mL Streptomycin in an incubator at 37 °C with 5% CO_2_. Myoblasts were kept from reaching confluence to avoid differentiation. Passage numbers were matched for controls and patient cells for the respective experiments, throughout all experiments passage numbers 8 to 10 have been used. Coverslips for myoblast immunohistochemistry were fixed at about 60–70% confluence. For differentiation confluent myoblasts were cultivated for 7 days in DMEM containing 5% horse serum.

### 4.4. Enzymes

Alglucosidase alfa (Myozyme^®^, PZN: 04796579) was obtained from Sanofi Genzyme, Cambridge, Massachusetts, USA.

### 4.5. Moss-GAA Expression Strain Construction

DNA sequence of human lysosomal alpha-glucosidase gene (GAA, NCBI Reference: NM_000152.5) starting from bp 511 encoding for a 110 kDa precursor protein was chosen to ensure correct intracellular trafficking to and processing in the lysosomes [30]. The construct was synthesized (GeneArt, Thermo Fisher Scientific, Regensburg, Germany) in frame with a plant secretory signal peptide sequence and sub-cloned into a moss expression vector. Linear expression cassettes were obtained from the plasmids by restriction. Expression cassettes comprise plant regulatory sequences, plant secretory signal peptide sequence, partial human GAA gene sequence and neomycin-phosphotransferase gene (nptII) under control of cauliflower mosaic virus (CaMV) 35S promoter.

To generate moss cell lines producing the GnGn variant of moss-GAA, protoplasts of a moss double-knockout line devoid of plant specific α-1,3-fucose and β-1,2-xylose residues on its *N*-glycans [20] were transformed with the expression cassettes by a PEG-based method and were selected using antibiotic geneticin (G418). Drug-resistant moss plantlets were screened for total moss-GAA accumulation per biomass, and the strain with the highest expression was chosen as production strain.

To produce moss-GAA with increased content of terminal mannose (Man5 variant), the above strategy was applied using a recipient strain with additional *N*-acetylglucosaminyltransferase I gene knock-out [15].

### 4.6. Production and Purification of Moss-GAA

Cultivation of the moss-GAA producing strains (GnGn and Man5) was done as described in [15]. The moss was grown in a pure mineral culture medium without any antibiotics or animal-derived component. Because of the lack of C-terminal vacuolar signal, moss-GAA was efficiently secreted into the culture medium. At the end of cultivation, the culture broth was clarified and concentrated. Moss-GAA was purified to homogeneity (>95% via SE-HPLC) using hydrophobic interaction chromatography (Toyopearl Butyl 650-M, Tosoh; Bioscience GmbH, Griesheim, Germany) followed by ion-exchange chromatography (Toyopearl Sulfate-650F, Tosoh, Bioscience GmbH, Griesheim, Germany). Subsequently, buffer was exchanged to 25 mM Na-Phosphate pH 6.2 and moss-GAA was concentrated to approx. 1 mg/mL. Purified moss-GAA was stored at −80 °C until further use.

### 4.7. Preparation of Moss-GAA Man3 Variant

To produce the moss-GAA variant with terminal paucomannosidic glycans, the moss-GAA GnGn has been treated with β-*N*-Acetylglucosaminidase (NEB, P0744, Frankfurt, Germany), following the manufacturers protocol, to specifically cleave off the *N*-acetylglucosamine motifs.

### 4.8. Analysis of Moss-GAA Variants

#### SDS-Page and Western Blot

Samples were denatured in LDS sample buffer (NuPage, Life Technologies, Carlsbad, CA, USA) at 95 °C for 5 min (or at 50 °C for 5 min for western blot) in the presence of dithiothreitol (NuPage reducing agent, Life Technologies). NuPAGE Bis-Tris 4–12% (Life Technologies) were used for protein separation. Western blot was performed using Novex Semi-dry blotter (Life Technologies). Primary antibody used was: rabbit polyclonal antibody to human GAA (Abnova, Taipei, Taiwan). Secondary antibody: anti-rabbit HRP (abcam, Cambridge, UK).

Specific activity has been measured in an assay using para-nitrophenol-α-d-glucopyranoside as substrate. Briefly: 50 µL sample was mixed with 450 µL reaction buffer, containing 350 µL 25 mM Na-Citrat + 0.4% BSA, pH 4.0 and 100 µL 20 mM para-nitrophenol-α-d-glucopyranoside (Sigma Aldrich, Germany) and incubated for 60 min at 37 °C. Reaction was stopped with 1 mL 200 mM boric acid, pH 9.8. Absorbance at 410 nm was read in a 1 cm cuvette using SPECORD (Analytik Jena, Jena, Germany) spectrophotometer. Specific enzyme activity was calculated using millimolar extinction coefficient of 18.5 for para-nitrophenol and was expressed as µmol hydrolyzed para-nitrophenol-α-d-glucopyranoside per min per mg of GAA.

Glycan analysis of moss-GAA variants and alglucosidase alfa was performed using HILIC-HPLC-MS. In short, *N*-glycans were released from the protein enzymatically using PNGase F. After cleanup and desalting, isolated glycans were labeled using procainamide. Labeled glycans were separated on an Agilent AdvanceBio Glycan Mapping column (2.1 × 150 mm, 1.8 μm) with a Security Guard Ultra precolumn (Phenomenex; Aschaffenburg, Germany) on a Nexera X2 HPLC system with a RF-20Axs Fluorescence Detector, equipped with a semimicro flow cell (Shimadzu, Korneuburg, Austria). Solvent A consisted of 80 mM formic acid, buffered to pH 4.4 with ammonia, solvent B of 95% ACN in solvent A. The applied gradient started with an initial hold of solvent B at 99% for 8 min and a decrease to 57% B over 60 min, following 25% B in 2 min, at a flow rate of 0.4 mL min^−1^; the column oven was set to 45 °C and flow cell thermostat to 45 °C. Fluorescence was measured with wavelengths Ex/Em 310 nm and 370 nm for the procainamide label. 3 µL injection volume was used for HPLC-MS runs with a post-column flow splitter connected to an Bruker amaZon ion trap mass spectrometer (Bruker, Bremen, Germany). The ion trap was operated in the positive ion mode with DDA. Target mass was set to m/z 700 and a m/z range from 150 to 1600 in the enhanced resolution mode with 150,000 ICC target and 200 ms maximum accumulation time. Data was analyzed in postrun analysis with LabSolutions 5.73 (Shimadzu, Germany) and Bruker Daltonics DataAnalysis 4.0.

### 4.9. GAA Uptake Assay

Myoblasts were grown to 70% confluence, Myozyme^®^ (10 µg/mL)/moss-GAA variants (10 µg/mL each) and 25 mM Na-Phosphat-buffer pH 6.2 (enzyme diluent) as a negative control respectively was added the cells and incubated for 24 h in an incubator at 37 °C with 5% CO_2_.

### 4.10. Preparation of Cell Lysates

After 24 h of incubation the medium was aspirated, cells were washed with 1xPBS twice, trypsinized and collected via centrifugation at 1000 g at room temperature. The pellet was resolved in 1 mL of 1xPBS and cells were centrifuged for 10 min, at 16,100× *g*, 4 °C. PBS was aspirated completely and 100 µL of aqua ad injectabilie was added. After that ultra-sonification (Sonopuls ultrasonic homogenizer HD2070 with sonotrode MS73, Bandelin, Germany) was performed (settings: 30 s; power 40 s; 5 cycles). After 10 min centrifugation with 16,100× *g* at 4 °C, supernatant was collected in a fresh tube and protein concentration was determined via Nanodrop 1000; (ThermoFisher, Waltham, Massachusetts, USA). Concentration of cell lysate was adjusted to 4 µg/µL.

### 4.11. GAA Activity Assay

GAA activity assays were measured in triplicates on black 96 well plates using 20 µL of 4 µg/µL cell lysate/standards. 80 µL of reaction buffer (0.25 mM 4-Methylumbelliferyl alpha-d-Glucopyranoside, 56 mM citric acid, 88 mM Na_2_HPO_4_, 0.4% BSA) was added to the samples, mixed for 10 s at 900 rpm and incubated for 60 min at 37 °C. The reaction was stopped using stop buffer (0.1 M Glycin, 0.1 M NaOH). Measurement was performed using a Tecan Infinite plate reader with following settings: orbital mixing step for 5 s, amplitude 1.5 mm, excitation: 360 nm, emission: 450 nm, gain: 50.

### 4.12. Real Time Metabolic Measurements

Metabolic measurements were performed using the Seahorse XFp Extracellular Flux Analyzer (Seahorse Bioscience; North Billerica, Billerica, MA, USA). For this, myoblasts were seeded in XFp Cell Culture Miniplates (103025-100, Seahorse Bioscience), at a density of 1.5 × 10^4^ cells per well and incubated overnight.

To investigate glycolytic function cells were incubated in unbuffered Basal Assay Medium (Seahorse) supplemented with 1 mM glutamine pH 7.4 at 37 °C without CO_2_ for 1 h before the assay. Following sequential injection of glucose (10 mM), oligomycin (1.0 μM), and 2-deoxy-d-glucose (50 mM) extracellular acidification rate (ECAR) was measured. Analysis of the data was performed well-wise.

### 4.13. PAS Staining

Cells grown on coverslips were fixed with 5% glacial acetic acid in 96% EtOH, rinsed for 1 min in slowly running tap water and immersed in periodic acid solution for 5 min at room temperature. In the next step coverslips were rinsed in several changes of distilled water and immersed in Schiff’s reagent for 15 min at room temperature. Following another wash step in running tap water for 5 min cells were counterstained in hematoxylin solution (for 90 s) and rinsed in running tap water for 5 min. Finally, the coverslips were mounted using fluorescent mounting medium (DAKO, Germany).

### 4.14. Immunohistochemistry

Cells were fixed with methanol (−20 °C). Primary antibodies used for staining include: GAA (ab166781, abcam, Cambridge, UK) LAMP1 (ab24170, abcam) and secondary antibodies Alexa Fluor 488 donkey anti-mouse AffiniPure (715-545-151, Jackson ImmunoResearch; West Grove, Pennsylvania, USA) and Alexa Fluor 594 goat anti-rabbit (A11012, Invitrogen, Carlsbad, California, USA). DNA was visualized with DAPI (4,6-diamidino-2 phenylindole, dihydrochloride) containing mounting medium (Vector Laboratories Burlingame, CA, USA, H-1200).

### 4.15. Microscopy and Image Analysis

All images were obtained using an Olympus FluoView FV1000/BX 61microscope (Shinjuku, Tokio, Japan) equipped with a 1.42 NA 60× objective and 3× zoom magnification. Image analysis was performed using ImageJ software (Wayne Rasband, USA).

### 4.16. Statistical Analysis

Statistical significance was determined by two-tailed Student’s *t*-test; error bars represent SE. * *p* ≤ 0.05 was considered statistically significant. ** indicates *p* values < 0.01; *** indicates *p* values < 0.001, and **** indicates *p* values < 0.0001.

## Figures and Tables

**Figure 1 ijms-21-02642-f001:**
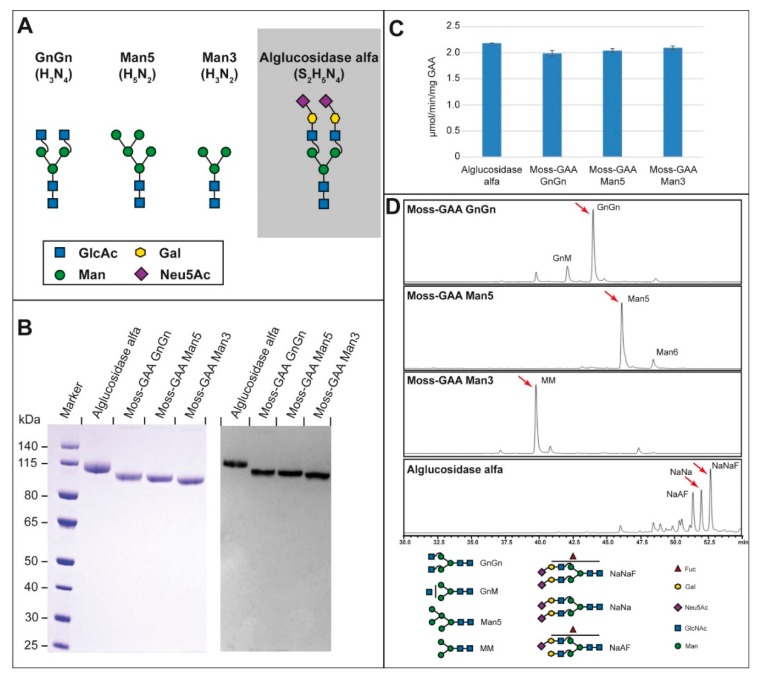
Characterization of moss-GAA variants in comparison to alglucosidase alfa: (**A**) modifications (GlcAc = *N*-Acetylglucosamine; Gal = Galactose; Man = Mannose; Neu5Ac = *N*-Acetylneuraminic acid), (**B**) reducing SDS-PAGE with Coomassie staining (left panel) and western blot (right panel), (**C**) specific enzyme activity, (**D**) total glycan profile via HILIC-HPLC. The red arrows in (**D**) correspond to the glycan structures in (**A**). The three major peaks are NaNaF, NaNa, and NaAF.

**Figure 2 ijms-21-02642-f002:**
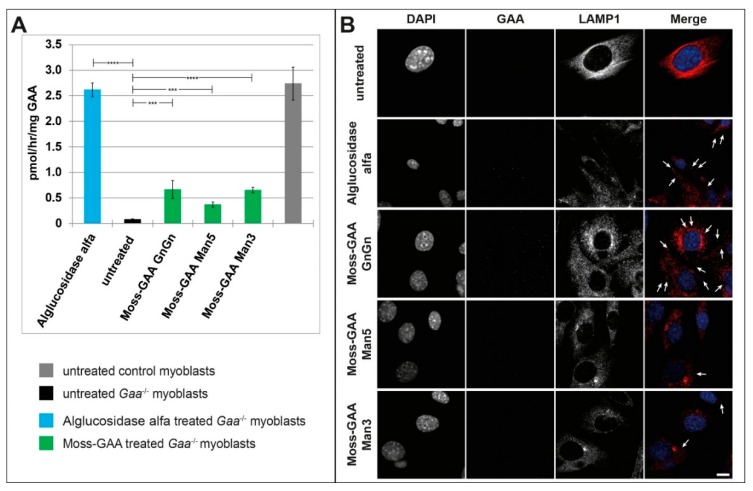
Uptake and activity of rhGAA in immortalized *Gaa*^−/−^ mouse myoblasts. (**A**) GAA activity assay of *Gaa*^−/−^ mouse myoblasts before and after treatment with all tested rhGAA variants. GAA activity is displayed in pmol/h/mg (total protein). (**B**) Immunofluorescence staining of *Gaa*^−/−^ mouse myoblasts for GAA and the lysosomal marker LAMP1 before and after treatment with all tested rhGAA variants. Scale bar 10 µm.

**Figure 3 ijms-21-02642-f003:**
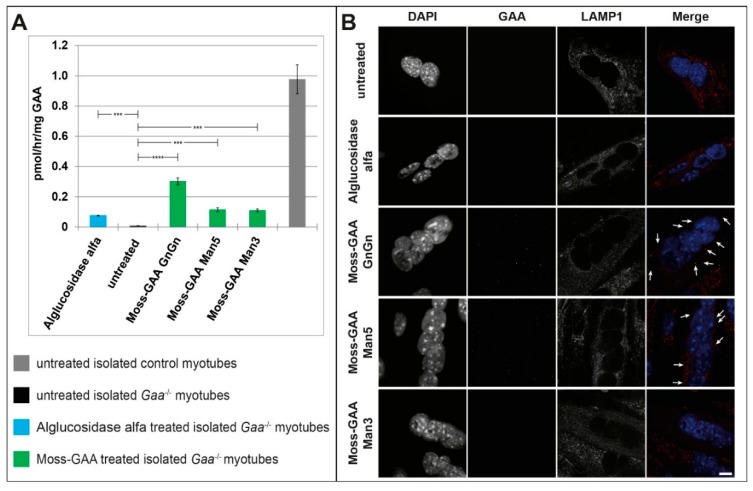
Uptake and activity of rhGAA in immortalized *Gaa*^−/−^ mouse myotubes. (**A**) GAA activity assay of isolated *Gaa*^−/−^ mouse myotubes before and after treatment with all tested rhGAA variants. GAA activity is displayed in pmol/h/mg (total protein). (**B**) Immunofluorescence staining of *Gaa*^−/−^ mouse myotubes for GAA and the lysosomal marker LAMP1 before and after treatment with all tested rhGAA variants. Scale bar 10 µm.

**Figure 4 ijms-21-02642-f004:**
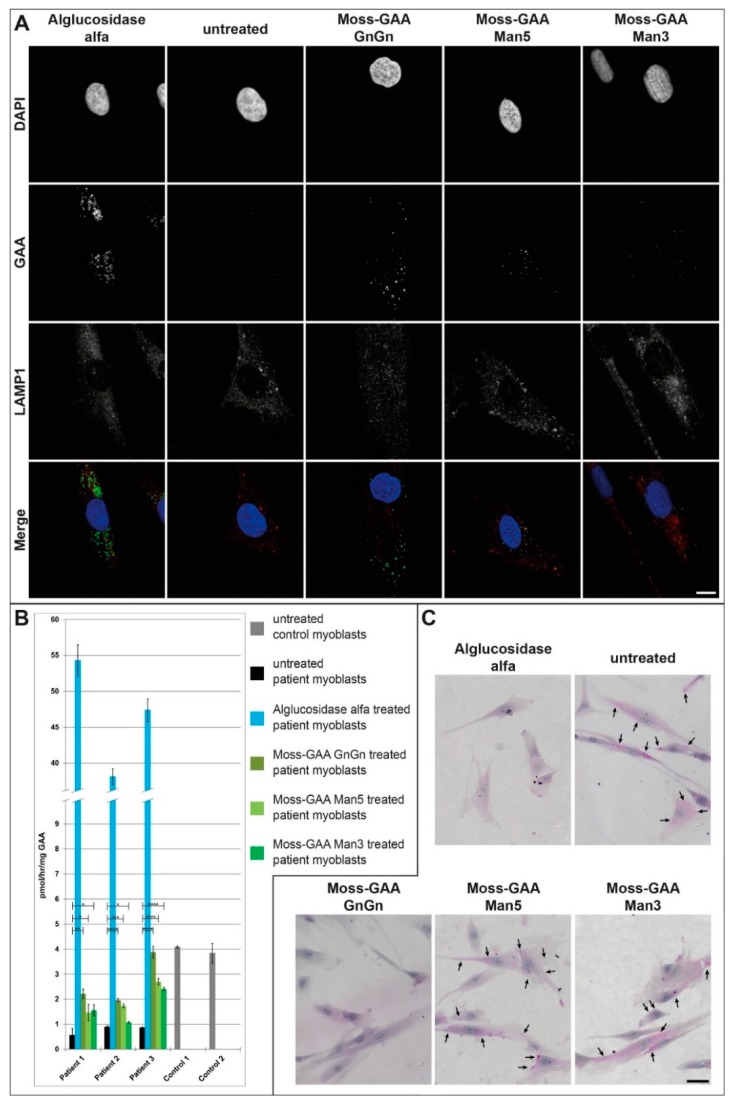
Uptake and activity of rhGAA in human myoblasts. (**A**) Immunofluorescence staining of Pompe patient primary myoblasts (example patient-1) for GAA and the lysosomal marker LAMP1 before and after treatment with all tested rhGAA variants. Scale bar 10 µm. (**B**) GAA activity assay of Pompe patient primary myoblasts before and after treatment with all tested rhGAA variants. GAA activity is displayed in pmol/h/mg (total protein). (**C**) PAS staining of Pompe patient primary myoblasts (example patient-1) before and after treatment with all tested rhGAA variants. Scale bar 40 µm.

**Figure 5 ijms-21-02642-f005:**
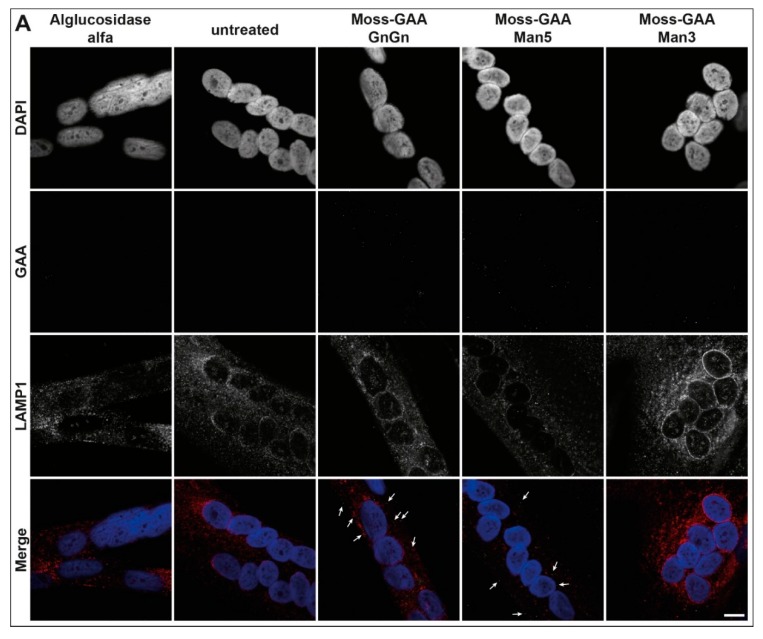
Uptake and activity of rhGAA in human myotubes. (**A**) Immunofluorescence staining of Pompe patient myotubes (example patient-1) for GAA and the lysosomal marker LAMP1 before and after treatment with all tested rhGAA variants. Scale bar 10 µm. (**B**) GAA activity assay of Pompe patient myotubes before and after treatment with all tested rhGAA variants. GAA activity is displayed in pmol/h/mg (total protein). (**C**) PAS staining of Pompe patient myotubes (example patient-1) before and after treatment with all tested rhGAA variants. Scale bar 40 µm.

**Figure 6 ijms-21-02642-f006:**
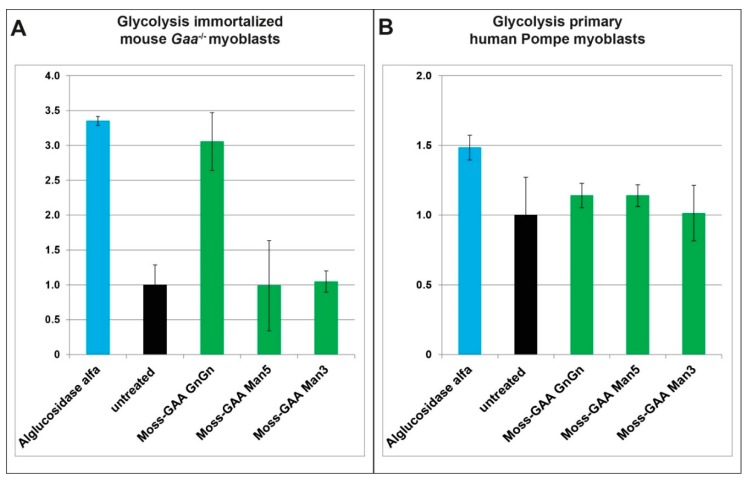
Glycolytic measurements. Measurement of glycolysis has been performed in (**A**) immortalized *Gaa*^−/−^ mouse myoblasts (normalized to untreated *Gaa*^−/−^ mouse myoblasts); and (**B**) Pompe patient primary myoblasts (normalized to untreated patient myoblasts).

**Table 1 ijms-21-02642-t001:** Characteristics of primary myoblast cultures used.

	*GAA*-Mutations	Sex	Age at Biopsy	Muscle Used
Patient-1	c.-32-13T>G/ c.1396delG	male	43	M. biceps brachii
Patient-2	c.-32-13T>G/ c.1942G>A	female	67	M. quadriceps femoris
Patient-3	c.-32-13T>G/ c.-32-13T>G	male	55	M. vastus lateralis
Patient-4	c.-32-13T>G/ c.1446delC	male	31	unknown
Control-1	---	female	49	M. vastus lateralis
Control-2	---	male	32	M. gastrocnemius

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
