# Peer review of "Moss-Derived Human Recombinant GAA Provides an Optimized Enzyme Uptake in Differentiated Human Muscle Cells of Pompe Disease"

_ijms, 2020, doi:10.3390/ijms21072642_

Round 1
Reviewer 1 Report
At this time I would recommend this manuscript be ACCEPTED WITH MAJOR CONCERNS. The study described in this manuscript is elegantly designed and well written, however the graphical representation of the data makes it difficult to review fully.
MAJOR CONCERNS
- All GAA enzyme activity graphs should be represented as they are in Figure 1 with the y-axis indicating “umol / min / mg [protein]” rather than each having an internal normalization. This will make all the activities cross comparable.
- In Figures 4 & 5 the goal is not to compare the rhGAAs within different patient cells it’s to compare the different rhGAAs to one another but these comparisons aren’t statistically done because they are represented on separate graphs.
- By combining these, it will also eliminate the need to graph the same data 3x as is done with the untreated control cells.
- Also it eliminates choosing one untreated control cell line to normalize to over the other as is done in Figure 4.
- It appears that the same data is represented in Figures 2 & 3 for untreated controls, even though Figure 2 is myoblasts, and Figure 3 is myotubes, it is hard to believe that they (appear) to have the exact same SE
- IF & PAS panels would benefit from a row of untreated normal controls.
MINOR CONCERNS
- Abbreviations should be written out at first introduction.
- Include the word “onset” when introducing “IOPD”
- There are several places where single sentences stand alone as a paragraph. These sentences should be added to the previous or following paragraph.
- “GAA-/-“ when referring to mouse genotype should be “Gaa-/-“
- Universally use “MossGAA” or “moss-GAA”
- “Patient 4” is used in section 2.5 but is not previously described.
- Line 192 is awkward because many modifiers are applied to several preceding words.
- On graphs y-axis needs to have periods for decimal place not commas.
- Are the representative images in Figure 2 all the same magnification? The nuclei diameters are drastically different.
- In Figure 5 – only Control 2 is used, why not also include control 1, which was used to normalize in Figure 4.
- Missing MossGAA-Man3 graph in Figure 5B
- Missing y-axis unit for Figure 6
Author Response
We thank indeed reviewer 1 for the helpful suggestions and comments to improve our submission. Here our point by point feedback:
At this time I would recommend this manuscript be ACCEPTED WITH MAJOR CONCERNS. The study described in this manuscript is elegantly designed and well written, however the graphical representation of the data makes it difficult to review fully.
MAJOR CONCERNS
- All GAA enzyme activity graphs should be represented as they are in Figure 1 with the y-axis indicating “umol / min / mg [protein]” rather than each having an internal normalization. This will make all the activities cross comparable.
We changed the figures 2, 3, 4 and 5 accordingly. Instead of normalizing to an internal control values are now presented as pmol/hr/mg [total protein].
- In Figures 4 & 5 the goal is not to compare the rhGAAs within different patient cells it’s to compare the different rhGAAs to one another but these comparisons aren’t statistically done because they are represented on separate graphs.
- By combining these, it will also eliminate the need to graph the same data 3x as is done with the untreated control cells.
- Also it eliminates choosing one untreated control cell line to normalize to over the other as is done in Figure 4.
We fused the graphs for the different enzymes in figures 4 and 5.
- It appears that the same data is represented in Figures 2 & 3 for untreated controls, even though Figure 2 is myoblasts, and Figure 3 is myotubes, it is hard to believe that they (appear) to have the exact same SE
The SE between figure 2 and 3 looks very similar, but is actually different. In the figure normalized to controls (before revision) it was 0.10 for figure 2 and 0.11 for figure 3 – after revision with values displayed as GAA activity in pmol/hr/mg of total protein it is 0.32 for figure 2 and 0.10 for figure 3.
- IF & PAS panels would benefit from a row of untreated normal controls.
We do agree that it would be nice to show more conditions. However, we fear that due to the fact that the panels are already (quite full) big adding more panels would make it too crowded for readers and the core information will be lost. Furthermore, the pathomechanism for Pompe disease is well described and this wouldn’t add to the core information, thus we think it is not necessary to add these panels.
MINOR CONCERNS
- Abbreviations should be written out at first introduction.
We added the full spelling for LSD (line 49), and ERT (line 26).
- Include the word “onset” when introducing “IOPD”
We included “onset” at the introduction of IOPD (line 36).
- There are several places where single sentences stand alone as a paragraph. These sentences should be added to the previous or following paragraph.
We fused paragraphs in following lines: 107, 125, 145, 180, 198, 293, 382, 394.
- “GAA-/-“ when referring to mouse genotype should be “Gaa-/-“
We changed this throughout the text and in figures 2, 3 and 6, thanks for noticing.
- Universally use “MossGAA” or “moss-GAA”
We changed this within in the text (lines 28, 146, 363) as well as in figures 2-6 to moss-GAA.
- “Patient 4” is used in section 2.5 but is not previously described.
We included the description for this patient in table 1.
- Line 192 is awkward because many modifiers are applied to several preceding words.
We deleted “that there is” to make the sentence easier to follow.
- On graphs y-axis needs to have periods for decimal place not commas.
This was changed in figures 3 and 5.
- Are the representative images in Figure 2 all the same magnification? The nuclei diameters are drastically different.
Yes, figures are all the same magnification and represent a range of different sized nuclei found in these cell cultures.
- In Figure 5 – only Control 2 is used, why not also include control 1, which was used to normalize in Figure 4.
This is correct. Due to the separation of differentiated myotubes from undifferentiated cells there is a lot of material necessary to get the necessary amount for measurement. Primary muscle cell lines vary in their differentiation efficiency; thus we could only include one control and did also use a cell line from patient 4 instead of patient 3.
- Missing MossGAA-Man3 graph in Figure 5B
We included the graph for moss-GAA Man3 in figure 5B now.
- Missing y-axis unit for Figure 6
Figure 6 shows the glycolysis of treated cells normalized to untreated cells. Due to the fact that we normalized and thus set untreated cells to 1 there is no unit.
Reviewer 2 Report
The paper by Hintze and colleagues presents data from experiments, based on moss derived recombinant GAA in the treatment of muscle glycogen deposits resulting from Pompe disease. The paper is well written, and their claims is adequately supported by several experiments they have undertaken. They highlight the importance of targeting, which may be a major factor-as traditionally the production relied on mammalian cell lines, for the 'requisite' pattern of glycosylation. One issue not addressed, which may be beyond the scope of the current paper is the issue of antigenicity/immunogenicity; which may be a factor limiting response to currently available commercial enzymes.
Author Response
We thank indeed reviewer 2 for the helpful suggestions and comments. Here our point by point feedback.
The paper by Hintze and colleagues presents data from experiments, based on moss derived recombinant GAA in the treatment of muscle glycogen deposits resulting from Pompe disease. The paper is well written, and their claims is adequately supported by several experiments they have undertaken. They highlight the importance of targeting, which may be a major factor-as traditionally the production relied on mammalian cell lines, for the 'requisite' pattern of glycosylation. One issue not addressed, which may be beyond the scope of the current paper is the issue of antigenicity/immunogenicity; which may be a factor limiting response to currently available commercial enzymes.
We thank the reviewer for this positive evaluation.
Round 2
Reviewer 1 Report
The authors have addressed the majority of my concerns but still not addressed some of the minor concerns:
- “GAA-/-“ when referring to mouse genotype should be “Gaa-/-"; GAA should be used for human gene and GAA used for human and mouse GAA protein
- Universally use “MossGAA” or “moss-GAA”
Author Response
We thank the reviewer again for the comments:
So we changed the lines 100,102,110,111,1113,117,121,129,130,132,193,195,203,218,272, as well the figures 2,3,6, to GAA-/-
Gaa was changed to GAA in lines 112,131,164, and 187
We did already use moss-GAA universally, we re-checked this again.
Reviewer 2 Report
Manuscript has had minor revision, which does not substantially alter the claims, supported by experimental evidence. Perhaps, a line in their discussion to state, which may already be implied, that at the organismal level there may be other tissues/organs that could take up their modified moss-derived GAA, which could ultimately impact on delivery to muscle tissue. Incidentally, Gaa in their revision should be changed to GAA.
Author Response
We thank reviewer 2 for the suggestion.
Therefore, we modified the sentence "It remains to be shown whether our results can be confrirmed in the Pompe mouse model." to " Considering the organismal level, it remains to be shown whether our results can be confirmed in the Pompe mouse model or other tissue/organs could take up moss-GAA and thus impact on muscle delivery.
Gaa was changed to GAA in lines 112,131,164, and 187.